# Dual Topoisomerase Inhibitor Is Highly Potent and Improves Antitumor Response to Radiotherapy in Cervical Carcinoma [note 1]

**DOI:** 10.3390/ijms26072829

**Published:** 2025-03-21

**Authors:** Inken Flörkemeier, Hannah L. Hotze, Anna Lena Heyne, Jonas Hildebrandt, Jörg P. Weimer, Nina Hedemann, Christoph Rogmans, David Holthaus, Frank-André Siebert, Markus Hirt, Robert Polten, Michael Morgan, Rüdiger Klapdor, Axel Schambach, Astrid Dempfle, Nicolai Maass, Marion T. van Mackelenbergh, Bernd Clement, Dirk O. Bauerschlag

**Affiliations:** 1Department of Gynecology and Obstetrics, Kiel University and University Medical Center Schleswig-Holstein, Campus Kiel, 24105 Kiel, Germanydirk.bauerschlag@med.uni-jena.de (D.O.B.); 2Pharmaceutical Institute, Department of Pharmaceutical and Medicinal Chemistry, Christian-Albrecht University of Kiel, 24118 Kiel, Germany; 3Priority Research Area Kiel Nano, Surface and Interface Sciences (KiNSIS), Kiel University, 24118 Kiel, Germany; 4Clinic of Radiotherapy, Kiel University and University Medical Center Schleswig-Holstein, Campus Kiel, 24105 Kiel, Germany; 5Institute of Experimental Hematology, Hannover Medical School, 30625 Hannover, Germany; 6Department of Gynecology and Obstetrics, Hannover Medical School, 30625 Hannover, Germany; 7Department of Gynecology and Obstetrics, Albertinen Hospital Hamburg, 22457 Hamburg, Germany; 8Division of Hematology/Oncology, Boston Children’s Hospital, Harvard Medical School, Boston, MA 02115, USA; 9Institute of Medical Informatics and Statistics, Kiel University and University Medical Center Schleswig-Holstein, Campus Kiel, 24105 Kiel, Germany; 10Department of Gynecology, Jena University Hospital, 07747 Jena, Germany

**Keywords:** cervical cancer, dual topoisomerase inhibitor, chemoradiotherapy

## Abstract

Despite advances in vaccination and early detection, the total number of cases and deaths from cervical cancer has risen steadily in recent decades, making it the fourth most common type of cancer in women worldwide. Low-income countries in particular struggle with limited resources and treatment limitations for cervical cancer. Thus, effective medicines that are simple to manufacture are needed. The newly developed dual topoisomerase inhibitor P8-D6, with its outstanding ability to induce apoptosis, could be a promising option. In this study, the efficacy of P8-D6 in combination with radiochemotherapy against cervical carcinoma was investigated in established cell lines and in a translational approach in ex vivo patient cells by measuring the cytotoxicity, cell viability and caspase activity in vitro in 2D and 3D cell cultures. Treatment with P8-D6 resulted in significantly greater cytotoxicity and apoptosis induction compared to standard therapeutic cisplatin in both 2D and 3D cell cultures. Specifically, a considerably stronger anti-proliferative effect was observed. The treatment also led to morphological changes and a loss of membrane integrity in the 3D spheroids. Radiotherapy also benefited greatly from P8-D6 treatment. In fact, P8-D6 was a more potent radiosensitizer than cisplatin. Simple synthesis, favorable physicochemical properties and high potency make P8-D6 a promising cervical cancer drug candidate.

## 1. Introduction

Cervical carcinoma is the fourth most common cancer among women, with around 661,000 new cases and 348,000 deaths worldwide in 2022, predominantly in low- and middle-income countries [1]. This reflects major inequalities due to a lack of access to HPV vaccination, cancer screening and treatment, as well as social and economic factors [2]. In over 90% of the cases, it is caused by persistent infection with high risk human papillomavirus (hrHPV) [3]. To improve the prognosis for affected women, early detection and treatment are crucial, as the one-year survival rate for patients with advanced or recurrent cervical carcinoma is less than 30% [4,5]. The combination of chemotherapy and radiotherapy plays a central role in the treatment of cervical carcinoma, especially in the advanced stages of the disease and is mostly given as a combination of external beam radiotherapy and brachytherapy [6,7,8]. Cisplatin is commonly added to radiotherapy in the clinic. Cisplatin functions by penetrating the cells via different pathways and forming several DNA-platinum adducts initiating apoptosis [9]. However, platinum-based therapy is an intensive treatment, is associated with severe side effects and induces drug resistance that compromises the efficacy [5,10]. Although new treatment options or strategies have emerged, i.e., the Interlace or Keynote A18 study, successful therapy in cervical carcinoma remains a challenge [11,12].

The combination of cisplatin with the topoisomerase-I inhibitor topotecan improved overall survival in the metastatic setting [13,14]. Since topoisomerase inhibitors have proven to be effective in recurrent cervical cancer, treatment with a new dual topoisomerase inhibitor appears to be a promising therapeutic approach. The newly developed P8-D6 is a strong inducer of apoptosis by acting as a dual topoisomerase I/II poison [15]. This aza-analogous benzo[*c*]phenanthridine covalently stabilizes the enzyme-cleaved DNA complex of both topoisomerases (topoisomerase I/II) [15,16]. Topoisomerase regulates torsional stress in DNA by inducing single- and double-strand breaks [17]. This function is important for unwinding DNA to enable essential genome functions such as transcription, replication or recombination [18,19,20]. Thus, topoisomerases play a crucial role in managing the topology of DNA replication and chromosome condensation. The stabilization of topoisomerase I/II-DNA complexes causes an increased number of DNA strand breaks, leading to unstable DNA replication and apoptotic cell death [21]. Furthermore, the simultaneous inhibition of both topoisomerases contributes to the reduced development of resistance and offers significant advantages in therapy due to the increased efficiency [22,23]. Due to its important function in proliferation processes, this enzyme class is an important target in cancer therapy. P8-D6 was highly effective for various tumor entities in the NCI-60 human tumor cell line screen with a GI_50_ of 49 nM [15,24]. In gynecological tumors, such as ovarian cancer, P8-D6 induced a significantly higher rate of apoptosis compared to standard therapeutics [24,25,26]. Moreover, it is easy to synthesize in a two-step process and has extraordinary solubility and stability properties (Synthesis Appendix A) [15].

The efficacy of radiotherapy as a monotherapy is limited by radiotoxicity to normal tissue and by tumor cell resistance [27]. For this reason, the combination of radiotherapy and chemotherapy with so-called radiosensitizers is used for treatment. Cisplatin has these radiosensitizing properties. Radiosensitizers increase the sensitivity of cancer cells to radiotherapy by enhancing the effect of the radiation, thereby increasing DNA damage in tumors cells while sparing normal tissue. The following mechanisms of action are conceivable: (i) the DNA repair inhibition, (ii) oxygen enhancement, (iii) cell cycle modulation and (iv) formation of free radicals [17,28]. For example, cisplatin inhibits the repair of radiation-induced DNA damage and forms cross-links in the DNA, leading to the formation of adducts that hinder DNA replication and transcription and induces cell cycle arrest, particularly in the G2/M phase [29]. The inhibition of topoisomerases thus prevents proper DNA repair and leads to increased radiation-induced DNA damage, resulting in increased cell death [30,31]. This is why topoisomerase inhibitors are already used in combination with radiotherapy for various types of cancer, such as head and neck cancer, lung cancer or lymphomas [32,33,34,35]. Thus, combining radiosensitizers with radiation therapy can improve treatment outcomes for various cancers.

Preclinical models that closely mimic human tumors are essential for the understanding of the biological mechanisms of carcinogenesis and response to therapy. The study of viability and apoptosis in two-dimensional (2D) cell culture models is the gold standard in oncology research to determine the effect of drugs [36,37,38]. However, these models lack the typical tumor architecture, including cell–cell interactions in the third dimension and the tumor microenvironment (TME). The simple and efficient three-dimensional (3D) spheroid models better represent the dynamic tumor architecture, including typical features such as naturally occurring hypoxic regions and cell–cell interactions [39,40,41]. These models are also lower in cost, more streamlined and allow for easier evaluation of the product performance compared to in vivo therapy [41]. This makes them valuable tools in cancer research, i.e., for the validation of 2D cell culture results. The use of primary ex vivo cells can further improve the translational value of such approaches by considering the biological relevance, inter-patient variability and reduced selection effect at an earlier stage of drug discovery [41].

The aim of this study was to evaluate the efficacy of the new dual topoisomerase inhibitor P8-D6 in cervical carcinoma to potentially develop new therapeutic approaches for monotherapy and radiochemotherapy.

## 2. Results

In order to develop a new drug for the treatment of a cancer disease, preclinical studies need to investigate the in vitro potential of a compound. Cervical carcinoma cells are special because almost all tumors are caused by hrHPV infection. SiHa and CaSki cells are HPV-related carcinoma cells, while C-33A is HPV-negative [42] (Table 1).

### 2.1. P8-D6 Induces Apoptosis in 2D Cervical Carcinoma Cells

To investigate whether P8-D6 can lead to cell death in cervical carcinoma cells with more efficiency than the established therapeutic cisplatin, we analyzed the viability and apoptosis rate of several cervical carcinoma cell lines after 48 h of incubation with P8-D6 or cisplatin. Parameters such as the IC_50_ are often used to measure drug sensitivity. The viability analysis indicated that P8-D6 had stronger anti-proliferative properties against SiHa (IC_50_ = 2.15 µM) than cisplatin (IC_50_ = 17.12 µM) (Figure 1A). CaSki and C-33A cells were also more sensitive to P8-D6 than to cisplatin (Figure 1C,E). P8-D6 induced a significantly higher level of apoptosis in SiHa, CaSki and C-33A cells after 48 h than cisplatin and PBS (Figure 1B,D,F). Compared to topotecan, P8-D6 also induced significantly higher levels of apoptosis in SiHa cells (Appendix A). In terms of response, SiHa and CaSki were more resistant, while C-33A were more sensitive to P8-D6 and cisplatin therapy. In general, the HPV-independent tumors were more chemosensitive than the HPV-positive tumor cells. However, with an average 8.5-fold increase in efficacy compared to cisplatin therapy, P8-D6 showed a consistent improvement across all cell lines independent of HPV status.

Since it is known that both platinum-based treatment and topoisomerase inhibition lead to endogenous damage or replication stress, the formation of DNA double-strand breaks (DSB) and their homologous recombinational repair were evaluated using the biomarkers γH2AX and RAD51. The fluorescence images clearly demonstrate that P8-D6 induced a great amount of γH2AX and RAD51 foci in SiHa and CaSki cells, even at a low µM concentration (0.1 µM, (Figure 1G)). At 1 µM, P8-D6 was able to induce 3.63-fold more γH2AX foci in CaSki and 9.66-fold more in SiHa cells compared to cisplatin.

### 2.2. P8-D6 Induces Strong Effects in 3D Target Tumor

Since multicellular 3D tumor spheroids, as in vitro cell culture models, represent the in vivo situation more realistically than 2D models, the cell lines were cultivated in special 3D spheroid-forming ultra-low attachment plates. After spheroid formation, treatment was continued for an additional 48 h. Based on the 2D model experiments mentioned above, further staining and microscopic analyses were performed on the treated 3D cervical carcinoma spheroids to evaluate treatment efficacy (Figure 2A). Treatment with P8-D6 led to a change in the spheroid surface morphology. Aggregates, such as those formed by C-33A or SiHa cells, had more condensed structures after treatment with P8-D6 (Figure 2B). CaSki cells, which formed round spheroids without treatment, showed more indentations and a ruffled surface with a loss of membrane integrity by treatment with P8-D6. A considerable reduction in size was also the result of the treatment.

The P8-D6 treated spheroids almost exclusively showed a necrotic core and an intensive propidium iodide (PI) signal (Figure 2C–E). Different sensitivities with respect to P8-D6 treatment could be observed. Treating C-33A cells with 1 µM P8-D6 almost completely eliminated the spheroid (Figure 2E). Furthermore, the cytotoxicity to the cervical carcinoma spheroids was increased by P8-D6 treatment compared to cisplatin or PBS. In SiHa and C-33A spheroids, fluorescence signals were significantly higher after 48 h, while CaSki spheroids showed a large necrotic core even without treatment (Figure 3). P8-D6 treatment reduced viability by >18-fold compared to cisplatin (Figure 4). P8-D6 also significantly increased apoptosis in cervical carcinoma spheroids.

### 2.3. P8-D6 Induces Strong Effects in Ex Vivo Primary Tumor Cells

There is evidence that established tumor cell lines do not adequately reflect the drug sensitivity and behavior of cancer cells in the clinic. Therefore, more reliable cancer models are currently being developed, and the primary material is used to mimic the heterogeneity and pathophysiology of patient tumors. For the present study, ex vivo primary cells were cultured in 2D monolayers and analyzed for their sensitivity to P8-D6 and cisplatin. With an IC_50_ value of 0.378 µM, primary cells were sensitive to P8-D6 and showed significant apoptosis rates compared to PBS (Figure 5A). At a concentration of 50 µM, P8-D6 showed a greater reduction in viability (Figure 5A). However, the induction of apoptosis was comparable to cisplatin in primary cells (Figure 5B). Similarly, treatment minimized the rate of proliferation in the translational approach (Figure 5C).

### 2.4. P8-D6 Sensitizes Cervical Carcinoma Cells to Radiotherapy in 3D Cell Culture

In order to determine whether dual topoisomerase inhibition increases the benefit of radiotherapy, cervical carcinoma spheroids were treated with three different radiation doses (2.5 Gy, 5 Gy or 10 Gy) and increasing concentrations of P8-D6 compared to untreated or cisplatin-treated spheroids. Comparison of the irradiated control spheroids with non-irradiated spheroids showed a loss of compact structure and membrane integrity in the SiHa spheroids by both fluorescence and scanning electron microscopy (SiHa: Figure 6A). C-33A and CaSki spheroids fused more at the surface and showed fewer rough structures (CaSki: Appendix A, C33-A: Appendix A). Evaluation of the combined effects of radiotherapy and P8-D6 treatment showed that the addition of P8-D6 treatment resulted in tumor size reduction in the SiHa and CaSki spheroids, a loss of spheroid structure in the C-33A spheroids and an increase in the PI signal when compared to cisplatin treatment (SiHa: Figure 6A, CaSki: Appendix A, C33-A: Appendix A). Treatment with P8-D6 significantly increased apoptosis induction in unirradiated and irradiated spheroids compared to controls (SiHa: Figure 6B, CaSki: Appendix A, C-33A: Appendix A). Radiotherapy also improved the outcome of the treatment with cytostatic drugs, but only to a limited extent.

The fraction affected (fa)-values of the combined treatment were interpreted by the CompuSyn 1.0 software (ComboSyn Inc.; Cambridge, UK) to assess the combination index (CI) in the cervical carcinoma 3D spheroids, reflecting synergistic outcomes (SiHa P8-D6: Figure 6C,D, SiHa cisplatin: Appendix A, CaSki: Appendix A, C-33A: Appendix A). In terms of benefit, radiotherapy effects were clearly enhanced by P8-D6 treatment, but also to a lower extent from cisplatin therapy. Certain concentrations of P8-D6 and cisplatin can be reduced by radiotherapy. The results also indicated that the combination therapy is of greater benefit for SiHa and C-33A cells than for CaSki spheroids.

## 3. Discussion

Globally, cervical carcinoma continues to be a major threat to the health of women, particularly those living in regions with lower levels of the Human Development Index (HDI) [1]. In addition to vaccination, the addition of cisplatin to radiotherapy has led to major advances in the treatment of cervical carcinoma patients in recent decades [6]. Although several clinical studies have been conducted on targeted molecular therapy, cisplatin is still the standard radiosensitizer used to date [28,43,44]. The development of new drugs and research into their functional effects are of high clinical importance. Therefore, the aim of the present study was to investigate the antitumor properties of the new dual topoisomerase inhibitor P8-D6 for the treatment of cervical carcinoma and its radiosensitizing properties in vitro.

Despite progress in locally advanced disease, there are only a few treatment options for women with advanced, recurrent or persistent cervical carcinoma. For example, combining cisplatin/topotecan or cisplatin/paclitaxel and bevacizumab improved survival in these patients (GOG 240 study) [45,46]. In addition to cisplatin, topotecan can also act as a radiosensitizer, due to its inhibition of topoisomerase I enzymes involved in the repair of radiation damage [14,47,48,49]. Topoisomerase I inhibitors (e.g., topotecan, irinotecan) are therefore already being used in the treatment of cervical carcinoma [14,50]. Several phase I, phase II and phase III studies, such as the GOG-9913 (NCT00054444) study with a treatment of topotecan, cisplatin and radiation or the GOG-0127W (NCT01266447) study with a combination of topotecan with the PARP inhibitor veliparib, have already been conducted to investigate topotecan clinically [47,51,52,53,54,55,56,57,58,59,60,61]. Similarly, irinotecan has been used in combination with cisplatin in some phase II trials [62,63,64,65,66,67,68,69,70,71]. In addition to the aforementioned drugs, checkpoint inhibitors such as pembrolizumab or cemiplimab are being studied intensively for the treatment of cervical carcinoma, with benefits in PD-L1-positive patients (KEYNOTE 158, KEYNOTE 826) [72,73,74,75]. New results from the Keynote A18 study, investigating the effect of adding the PD-L1 inhibitor pembrolizumab to cisplatin-based radiochemotherapy showed an improvement in progression-free survival [11]. Similar results were obtained by employing chemotherapy prior to radiochemotherapy, as shown in the interlace study [12]. Thus, the idea of combining different drugs is a promising approach to treat cervical cancer more efficiently.

P8-D6 belongs to the class of topoisomerase inhibitors which covalently stabilizing the topo-DNA complex of both enzymes (Topo I/II) and leads to double-stranded DNA breaks and cell death [15,16,24]. Moreover, the sensitivity of tumor cells to P8-D6 is believed to be positively correlated with topoisomerase activity [16,24,25,26]. Previous studies have shown that the HPV oncoproteins E6 and E7 upregulate topoisomerase 1 [76]. Our study demonstrated that treatment with P8-D6 reduced viability, promoted apoptosis and induced genetic instability in both established cervical carcinoma cell lines and primary cells. Thus, the anticancer activity of P8-D6 observed in cervical carcinoma cells was in line with the previous studies in malignancies like ovarian cancer cells, breast cancer and multiple myeloma [24,25,26,77,78]. However, the HPV-negative cell line was found to be the most sensitive cervical carcinoma cell to P8-D6. This was not consistent with the HPV-related upregulation of topoisomerase I. Comparing the mRNA expressions of the cell lines C-33A, CaSki and SiHa, topoisomerase I and II β expression was highest in C-33A cells, followed by SiHa and CaSki (Appendix A) [79]. This confirmed the correlation of the efficacy of P8-D6 with topoisomerase I and II β expression.

Due to its unique mechanism of action as a dual topoisomerase poison, P8-D6 has great potential to become an effective and successful drug in oncology. The simultaneous inhibition of both topoisomerases contributes to a lower development of resistance, e.g., by upregulating the target enzyme [22,23]. Previous studies even found that P8-D6 was superior to a combination of topo-I (topotecan) and topo-II (etoposide) inhibitors in inducing apoptosis [24]. In addition, P8-D6 is easy to synthesize in two steps and has excellent solubility and stability properties, which would be advantageous in regions with a lower HDI (Synthesis Appendix A) [15].

In oncology, new therapeutic options are first investigated using in vitro models before being tested in vivo. Appropriate cell culture models are essential to better understand the physiology of tumors and to predict the efficacy of new therapeutic approaches [39,40,41]. Thus, 3D cell culture models offer an advantage over conventionally used 2D models because they can more realistically reflect the physiological parameters of tumors [36,37,38]. For this reason, in this study, P8-D6 was tested not only in 2D but also in 3D spheroid cell culture models. Spheroid formation of the cervical carcinoma cell lines has been reported previously [80,81,82,83]. We were able to confirm that HPV16-infected CaSki cells form more compact spheroids compared to HPV-negative C33A [82]. Similar to other studies, SiHa cells formed flat, loose spheroids [42,80]. Treatment with P8-D6 resulted in a significantly higher cytotoxic effect and apoptosis induction than with standard therapeutics like cisplatin or topotecan in both 2D and 3D cell cultures. In particular, a much more pronounced anti-proliferative effect was observed at lower concentrations in comparison with the standards. (Figure 1 and Figure 4). The treatment also caused the spheroids to change morphologically and lose membrane integrity. In comparison with other studies, we found that the methods we used are valid and consistent. For example, if the sensitivity of C-33A cells to cisplatin is analyzed, these cells are most sensitive [42,84,85]. In terms of clinical dosing, radiochemotherapy with cisplatin can serve as a good reference point. Treatment recommendations for advanced cervical cancer include the use of chemotherapy and brachytherapy. Cisplatin is administered once a week at a dose of 40 mg/m^2^ for 5 cycles, plus radiotherapy at 1.8–2 Gy per fraction (4–6 cycles) for a total of 50 Gy [86]. Considering the radiation doses used in our study, the comparability of the single dose is evident. For cisplatin, there are studies that indicate 1 µM cisplatin as a clinically relevant dose [42,87,88,89].

Cisplatin is the standard treatment for patients with cervical carcinoma. However, resistance to cisplatin can develop, which continues to be a major challenge in treatment. Based on the mechanisms of resistance, there are several strategies to overcome resistance in cervical carcinoma: (i) the development of new agents, (ii) the improvement of drug formulation and (iii) a combinatorial approach of cisplatin with other agents [90,91]. The new compound P8-D6, which can also be used in combination with cisplatin, addresses two of these strategies and could thus contribute to improving the survival rate of patients with advanced or recurrent cervical carcinoma. In order to improve the survival rate of patients with cervical carcinoma, it is necessary to find treatment combinations with low toxicity and high efficacy [6,14]. Cisplatin and topotecan act as radiosensitizers [14,43,44,47,49]. For this reason, this study also investigated the effect of P8-D6 in combination with radiation on cervical carcinoma spheroids. The addition of P8-D6 to radiotherapy resulted in sensitization of the cells and a significant benefit in the treatment of cervical carcinoma, which was largely synergistic. For both cisplatin and P8-D6, the benefit of the combination was significantly greater in SiHa and C33-A cells than in CaSki cells. Overall, these data demonstrate a strong anti-tumor effect of P8-D6 under 2D and 3D cell culture conditions and provide further evidence for the dose reduction in radiation by co-treatment with the cytostatic agent P8-D6.

## 4. Materials and Methods

P8-D6 was synthesized as recently described [15] (Appendix A) and dissolved in PBS. Cisplatin and topotecan was obtained from the UKSH dispensary and dissolved in PBS.

### 4.1. Cell Preparation and Culture

The human papillomavirus-related cervical carcinoma cell lines SiHa (RRID: CVCL _0032) and CaSki (RRID: CVCL_1100), purchased from American Type Culture Collection (ATCC, Manassas, Virginia), were maintained in RPMI 1640 medium including L-glutamine, supplemented with 10% FBS and 60 IU(µg)/mL penicillin-streptomycin. The human papillomavirus-independent cervical carcinoma cell line C-33A (RRID: CVCL_1094) was cultured in DMEM, supplemented with 10% FBS and 60 IU(µg)/mL penicillin-streptomycin. Primary cervical carcinoma cells were isolated from cervical carcinoma patient tissue during surgery at Hannover Medical School (MHH) (Hannover, Germany). Subsequently, the tumor tissue was minced, and the primary cells were extracted from the tissue pieces by enzymatic digestion with Dispase II for 45 min at 37 °C. Afterwards, the cells were filtered through a 70 µm mesh and subsequently cultivated in RPMI 1640 medium, supplemented with 10% FBS, 100 U/mL penicillin, 100 µg/mL streptomycin and 1% sodium pyruvate. Cells were grown at 37 °C and 5% CO_2_ in a humidified incubator and subcultured at a confluency of 70–80%. Cell line authenticity was checked by STR profiling, as described previously [92], and routinely checked for Mycoplasma contamination using MycoAlert™ (Lonza, Basel, Schweiz). Informed consent was obtained from all donors. This study was conducted according to the guidelines of the Declaration of the Hannover Medical School and was approved by the Institutional Review Board of the Hannover Medical School, Hannover Ethics Commission (#6090) on 2 July 2018.

### 4.2. 2D Cell Culture

To evaluate cell viability and apoptosis in 2D cell culture, SiHa (5000 cells/well), CaSki (6000 cells/well), C-33A (10,000 cells/well) and primary (6000 cells/well) cells were seeded in a 96-well plate (Corning 3903, New York, NY, USA). The next day, the cell lines and patient-derived cells were treated in triplicate with P8-D6, cisplatin, topotecan or PBS for 48 h.

#### 4.2.1. 2D Viability and Apoptosis

ApoLive-Glo™ Multiplex Assay (Promega G6410, Fitchburg, WI, USA) was used as described in the manufacturer’s instructions (TM325) to measure viability and apoptosis in a single multiplexed assay. Viability was measured in fluorescence units (400_Ex_/505_Em_), and Caspase-Glo 3/7 cleavage was determined as relative luminescence units (RLU) using a microplate multimode reader (Spark, Tecan, Männedorf, Schweiz or GloMax^®^ Explorer, Promega, Fitchburg, WI, USA). For the calculation of relative apoptosis, caspase activity was divided by viability after the normalization of viability to control. Using viability data and logistic regression with four parameters, dose–response curves were plotted and inhibitory concentration 50% (IC_50_) values were calculated (GraphPad Software, Version 10.2.2, Boston, MA, USA). Growth monitoring of the patient-derived cells was analyzed over time using the automated cell imager CELLCYTE X™ (CYTENA, Freiburg, Germany) with the 4× objective. The total area of the cells was evaluated using the CELLCYTE Studio (CYTENA, Freiburg, Germany).

#### 4.2.2. Fluorescence Imaging (LSM)

To evaluate cellular DNA damage in 2D cell culture, SiHa (360 cells/well) and CaSki (1760 cells/well) cells were seeded in glass-bottomed 8-well chamber slides (Ibidi 80807, Gräfelfing, Germany). After seeding, cervical carcinoma cells were treated with P8-D6 [0.1 µM, 1 µM], cisplatin [1 µM] or PBS for 48 h. After treatment, the cells were washed with PBS, fixed with 4% paraformaldehyde for 10 min at room temperature and permeabilized with 0.2% Triton X-100 for 5 min at room temperature. After blocking with goat serum (1:20 in PBS) for 1 h at RT, cells were incubated with primary antibodies (anti-RAD51 1:100 (Proteintech 67024-1-Ig, Rosemont, IL, USA); anti-γH2A.X 1:100 (Cell Signaling 9718, Danvers, MA, USA)) at room temperature for 1.5 h. After washing with PBS, cells were incubated with secondary antibodies (Goat anti-Mouse IgG (H + L) (Alexa Fluor 594) 1:1000 (Thermo Fisher A-11032, Waltham, MA, USA), Goat Anti-Rabbit IgG (H + L) (Alexa Fluor^®^ 488) 1:1000 (Abcam ab150081, Cambridge, UK)) for 1.5 h at room temperature. Cells were washed with PBS and stained with DAPI/mounting medium (0.5 µg/mL) (Vectashield). A Zeiss LSM 880 microscope was used for fluorescence imaging (Carl Zeiss Microscopy, Jena, Germany). ZEN 2.5 software (Blue Edition, Carl Zeiss Microscopy, Jena, Germany) was used to evaluate the data.

### 4.3. 3D Cell Culture

#### 4.3.1. 3D Viability, Apoptosis and Cytotoxicity

To form spheroids, SiHa (5000 cells/well), CaSki (8000 cells/well) and C-33A (1000/well) cells were seeded into a 96-well ultra-low attachment plate (Corning 4520, New York, USA), centrifuged at 300× *g* for 1 min, and maintained for 24 h (CaSki) or 72 h (SiHa, C-33A). Part of the medium was removed before the spheroids were treated for a further 48 h with P8-D6, cisplatin or PBS and with or without radiotherapy (section Radiotherapy). CellTox™ Green Assay (Promega G8731, 0.04%, Fitchburg, WI, USA) was added concurrently with treatment. Growth monitoring and cytotoxicity was detected 24 h and 48 h after treatment using the automated cell imager NYONE^®^ Scientific (SYNENTEC, Elmshorn, Germany) with a 4× objective and the excitation sources and emission filters for bright field BF_Ex_/Green_Em_ (530/43 nm) and CellTox™ Green (Blue_Ex_ (475/28 nm)/Green_Em_ (530/43 nm)). The fluorescence data and images were evaluated using the YT-SOFTWARE^®^ (Version 23.04.27471, SYNENTEC, Elmshorn, Germany) using the Spheroid Quantification (2F) application. The application relates the spheroid mask in the bright field image to the average fluorescence intensity.

After treatment, spheroid viability and apoptosis were determined by RealTime-Glo™ (460_Em_) (Promega G9711, Fitchburg, WI, USA) and Caspase-Glo 3/7 (565_Em_) (Promega G8090, Fitchburg, WI, USA), using a microplate multimode reader (Spark, Tecan). The measurement was performed in a single multiplexed method according to the manufacturer’s instructions (TM431, TB323). For the calculation of relative apoptosis, caspase activity was divided by the viability after normalization of the viability to control. Using viability data and logistic regression with four parameters, dose–response curves were plotted and inhibitory concentration 50% (IC_50_) values were calculated (GraphPad Software Version 10.2.2, Boston, MA, USA).

For live/dead staining, cells were grown in ultra-low attachment plates and treated with P8-D6, cisplatin or PBS and with or without radiotherapy as described above. After treatment, part of the medium was removed and the spheroids were stained with propidium iodide (PI) (15 µM), calcein-AM (0.2 µM)) and Hoechst 33342 (1.8 µM) in medium for 3 h at 37 °C. For imaging, the automated cell imager NYONE^®^ Scientific (SYNENTEC, Elmshorn, Germany) with a 4× objective was used, and the excitation sources and emission filters for bright field BFEx/GreenEm (530/43 nm); Hoechst 33342 UVEx (377/50 nm)/BlueEm (452/45 nm); calcein-AM BlueEx (475/28 nm)/GreenEm (530/43 nm) and PI LimeEx (562/40 nm)/RedEm (628/32 nm) were applied. The images were evaluated using the YT-SOFTWARE^®^ (Version 23.04.27471, SYNENTEC, Elmshorn, Germany) using the Spheroid Quantification (2F) application.

#### 4.3.2. Scanning Electron Microscopy

The spheroids were grown in ultra-low attachment plates and treated with P8-D6, cisplatin or PBS and with or without radiotherapy, as described in the section “3D viability, apoptosis and cytotoxicity”. After treatment, the spheroids were fixed with 2.5% glutaraldehyde for 1 h at RT, washed with PBS and further re-fixed with 1% osmium tetroxide for 1.5 h at room temperature. The spheroids were washed again and subsequently dehydrated with an ascending ethanol series [25%, 50%, 75%, 96%, 100%] for 30 min. Using hexamethyldisilazane, the spheroids were transferred to charcoal stubs and air-dried overnight. For better conductivity, the spheroids were coated with gold particles and subsequently measured with scanning electron microscopy (Phenom XL, Phenom-world, Thermo Fisher, Darmstadt, Germany).

#### 4.3.3. Radiotherapy

Cells were irradiated with 6 MeV photons using a Siemens Artiste linear accelerator (Siemens Healthineers, Erlangen, Germany) at room temperature, with a 15 mm build up material (Polymethyl methacrylate) to ensure a homogeneous dose and a dose rate of 3 Gy/Min with a total dose of 2.5 Gy, 5 Gy or 10 Gy. The dose distribution was calculated using Eclipse v13.6 (Varian a Siemens Healthineers Company, Palo Alto, CA, USA) as the treatment planning software (TPS, Treatment Planning System). Radiotherapy was given immediately (less than 15 min) before drug treatment.

### 4.4. Combination Index and Dose Reduction Index

Data were analyzed using CompuSyn to identify synergistic drug combinations (ComboSyn Inc.; Cambridge, UK). For each combination of the drug and radiotherapy, as well as for monotherapy, the combination index (CI) and dose reduction index (DRI) values were determined. Therefore, viability was normalized to the negative control (PBS), then converted to fraction affected (fa)-values and analyzed with CompuSyn 1.0 software (ComboSyn Inc.; Cambridge, UK) [26,93,94]. The CI values indicate the effect of combining multiple therapies (synergistic, additive or antagonistic); the DRI values indicate the fold reduction in the dose of a therapy required in a combination to achieve the same efficacy (fa) as the single therapy. Using this approach, therapy combinations with CI values <  1 are synergistic, CI values = 1 are additive, CI values > 1 are antagonistic and DRI values  >  1 are favorable.

### 4.5. Statistical Analysis

Statistical tests were performed using GraphPad Prism 10 (GraphPad, Version 10.2.2, Boston, MA, USA). Gaussian distribution was tested using the Shapiro–Wilk normality test. Data from multiple groups were tested for statistically significant differences using one-way ANOVA. Statistically significant differences were assumed at *p*-values < 0.05 (*) according to Dunnett’s Test for multiple comparisons.

## 5. Conclusions

In summary, this study reports on the need for an effective treatment for cervical carcinoma and highlights new therapeutic strategies using a novel, effective drug. Due to its simple synthesis, favorable physicochemical properties and high efficacy, P8-D6 is a promising drug candidate against cervical carcinoma. Our study has shown that treatment with P8-D6 reduces viability, promotes apoptosis and induces genetic instability in both established cervical carcinoma cell lines and primary cells. Compared to radiotherapy or chemotherapy alone, the combination of a topoisomerase inhibitor and radiation offers effective advantages. This is the first study to demonstrate the anticancer activity of P8-D6 in cervical carcinoma and to propose the new drug P8-D6 as a radiosensitizer. However, in future studies, we aim to validate these results in vivo to confirm the role of P8-D6 in cervical carcinoma therapy and to test the robustness of our findings in a multi-organ system.

## 6. Patents

Clement, B.; Meier, C.; Heber, D.; Stenzel, L. novel pyrido [3,4-c] [1,9] phenanthroline and 11,12-dihydropyrido [3,4-c] [1,9] phenanthroline-derivatives and their use, in particular, for the treatment of cancer. PCT/EP2013/057212; granted for Europe, USA, Canada, Australia EP2834240, US9062054, CA 2869426, AU2013244918. Clement, B.; Meier, C.; Steinhauer, T.N. novel pyrido-phenanthroline derivatives, production and use thereof as medicaments. PCT/EP2017/080327; granted for Europe-EP3330270.

## Figures and Tables

**Figure 1 ijms-26-02829-f001:**
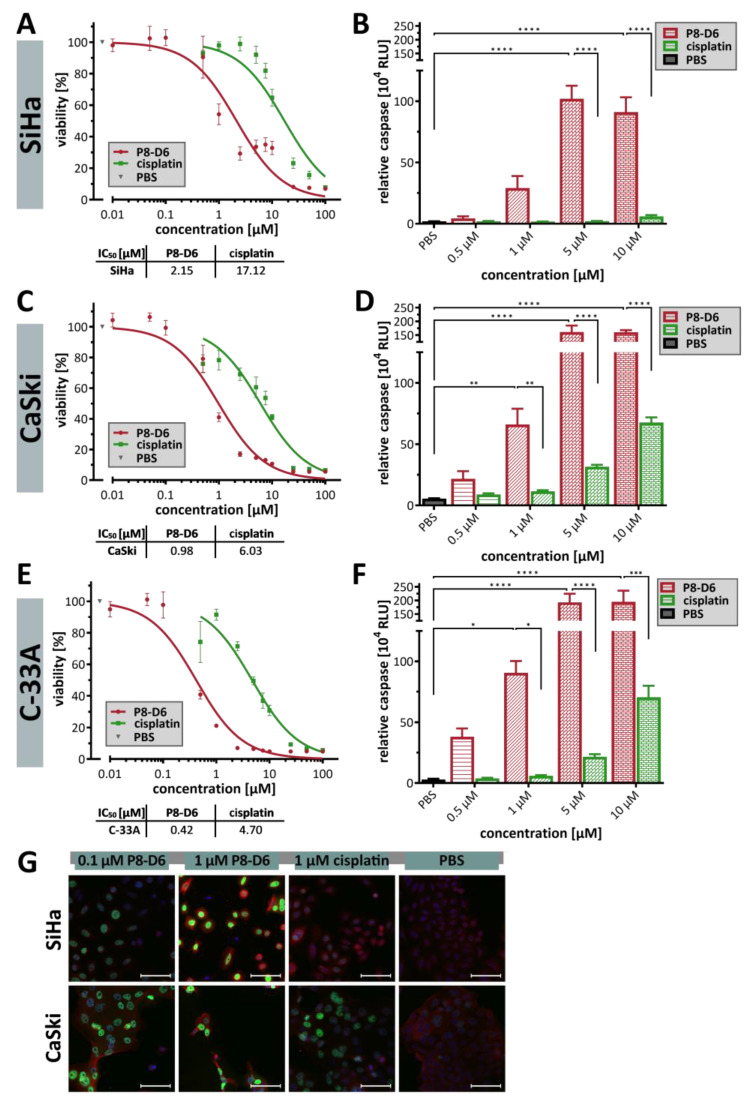
Antitumor responses to treatment with P8-D6 and cisplatin of the 2D cell lines of cervical carcinoma. SiHa, CaSki and C-33A cells were treated in a 2D monolayer cell culture with different concentrations of P8-D6, cisplatin and PBS in single therapy for 48 h. (**A**–**F**) Subsequently, the viability and caspase activity were determined. (**A**,**C**,**E**) The IC_50_ values were calculated using the viability data (SiHa, CaSki, C-33A). (**B**,**D**,**F**) The apoptosis rate is represented by the relative caspase activity (relative luminescence units, RLU). Data are means + SEM with analysis by one-way ANOVA, * (*p* < 0.05), ** (*p* < 0.01), *** (*p* < 0.001), **** (*p* < 0.0001). (**G**) Immunofluorescence images of SiHa and CaSki cells: DAPI-stained nuclei (blue), γH2A.X (green) and RAD51 (red) after treatment with cisplatin or P8-D6; scale bars, 100 μm.

**Figure 2 ijms-26-02829-f002:**
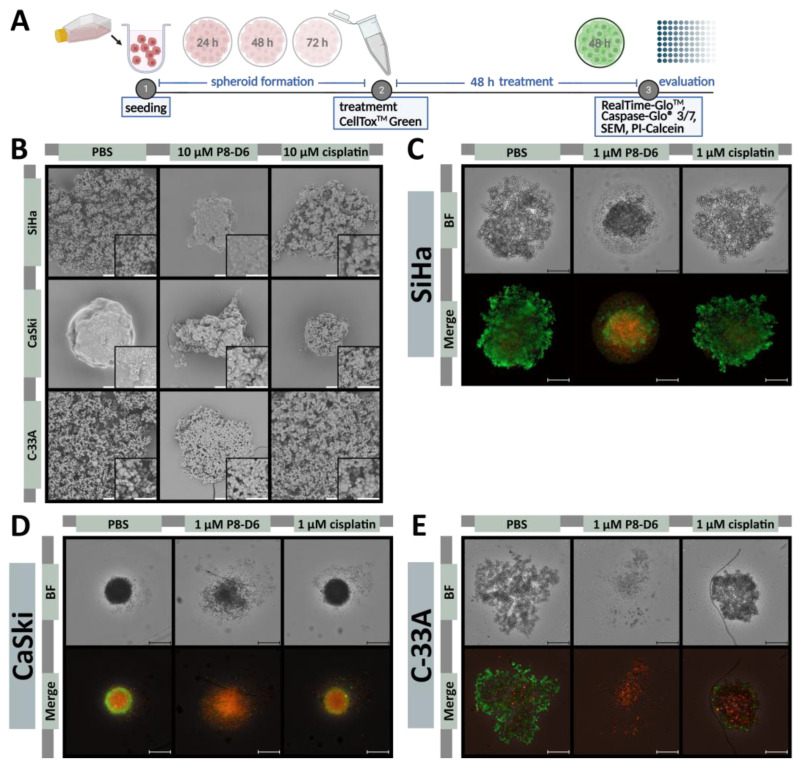
In vitro degradation and the evaluation of antitumor activity. Spheroids were maintained in ultra-low attachment plates for 72 h (SiHa and C-33A) or 24 h (CaSki) and subsequently treated with P8-D6, cisplatin and PBS in single therapy for 48 h as illustrated in the schedule (**A**). (**B**) Scanning electron microscopy images of the spheroids, which were treated with P8-D6, cisplatin or PBS; scale bars, 50 μm. SiHa (**C**), CaSki (**D**) and C-33A (**E**) spheroids were live/dead co-stained after treatment with PI (red), calcein-AM (green) and Hoechst 33342 (blue), and imaged by fluorescence microscopy; scale bars, 500 μm.

**Figure 3 ijms-26-02829-f003:**
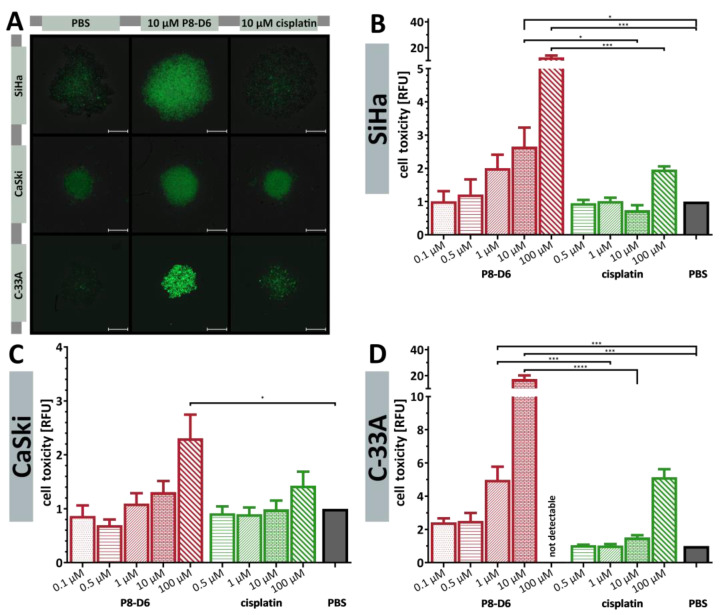
Toxicity of the dual topoisomerase inhibitor in 3D tumor spheroids. Spheroids were maintained in ultra-low attachment plates and treated as described in the schedule in Figure 2A. (**A**) After 48 h treatment, the cell toxicity was measured by fluorescence microscopy using CellTox™ Green (Promega, Fitchburg, WI, USA); scale bars, 500 μm. The fluorescence signals after treatment were quantified (relative fluorescence units, RFU) in the SiHa (**B**), CaSki (**C**) and C-33A (**D**) spheroids. Data are means + SEM with analysis by one-way ANOVA, * (*p* < 0.05), *** (*p* < 0.001), **** (*p* < 0.0001).

**Figure 4 ijms-26-02829-f004:**
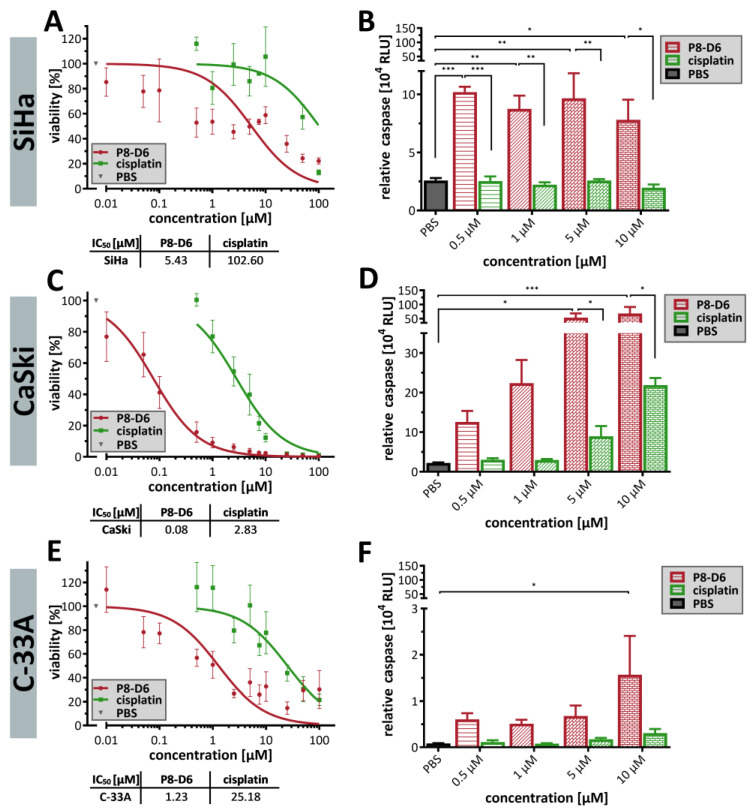
Antitumor responses to P8-D6 in 3D tumor spheroids. Spheroids were maintained in ultra-low attachment plates and treated as described in the schedule in Figure 2A. After 48 h treatment, the viability and caspase activity were measured. The IC50 values of P8-D6 and cisplatin were calculated using the viability data from the SiHa (**A**), CaSki (**C**) and C-33A (**E**) spheroids. Apoptosis is shown as relative caspase activity in the SiHa (**B**), CaSki (**D**) and C-33A (**F**) spheroids (relative luminescence units, RLU). Data are means + SEM with analysis by one-way ANOVA, * (*p* < 0.05), ** (*p* < 0.01), *** (*p* < 0.001).

**Figure 5 ijms-26-02829-f005:**
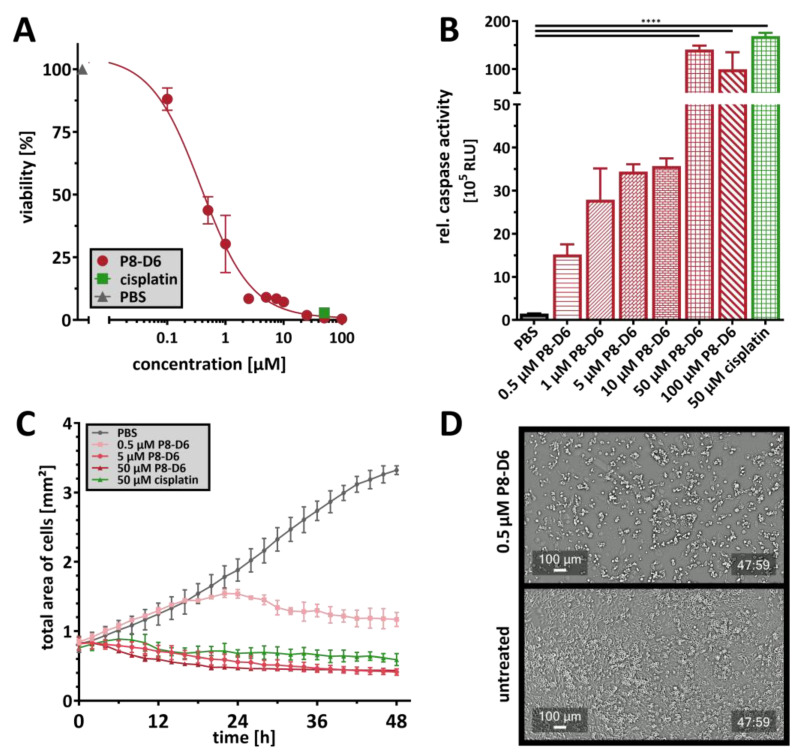
Antitumor effect of P8-D6 in primary cervical carcinoma cells. Primary cells were treated in a 2D monolayer cell culture with P8-D6, cisplatin and PBS in single therapy for 48 h. After treatment, the viability (**A**) and caspase activity (**B**) were measured. Data are means + SEM with analysis by one-way ANOVA, **** (*p* < 0.0001). (**C**) The total area of cells was determined over time. Data are means + SEM. (**D**) Cell growth videos showing the effects of treatment with P8-D6 after nearly 48 h compared to untreated controls; scale bars, 100 μm.

**Figure 6 ijms-26-02829-f006:**
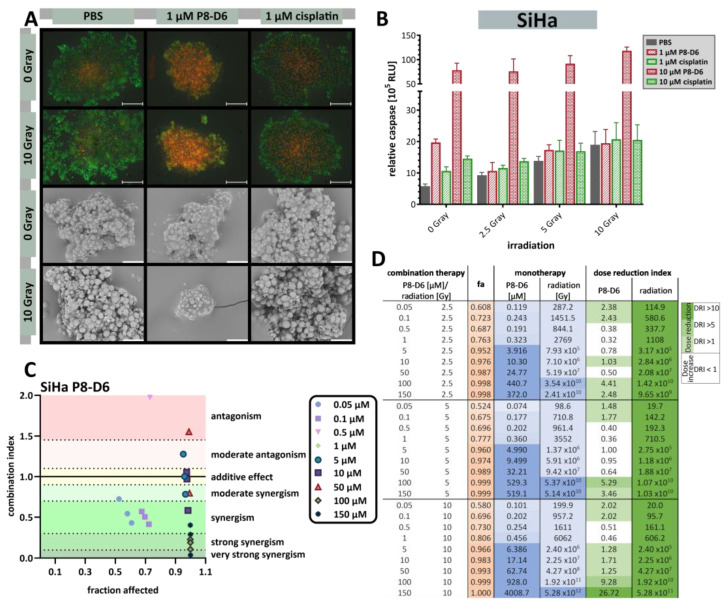
Antitumor responses to combined treatment with radiotherapy and P8-D6 in 3D SiHa cell culture models. SiHa spheroids were maintained in ultra-low attachment plates for 72 h, irradiated with a total dose of 2.5 Gy, 5 Gy or 10 Gy and immediately treated for 48 h with P8-D6, cisplatin or PBS. (**A, top**) After treatment, SiHa spheroids were live/dead co-stained with PI (red) and calcein-AM (green) and imaged by fluorescence microscopy; scale bars, 500 μm. (**A**, **bottom**) Scanning electron microscope images were taken of the spheroids treated with radiochemotherapy; scale bars, 50 μm. (**B**) Viability and caspase activity of the SiHa spheroids were measured after treatment (relative luminescence units, RLU). (**C**) Combination index (CI) for combining radiotherapy with P8-D6 was determined using the SiHa spheroids. CI values were calculated by CompuSyn 1.0 software (ComboSyn Inc.; Cambridge, UK) using the fraction affected (fa)-value. The fa-value represents the fraction of cell viability affected by therapy. fa color scalar (0 (white)-1 (light brown)) Combinations were considered synergistic when CIs were below 1.0. (**D**) The monotherapy column defines the concentrations that are needed in monotherapy to affect a certain fraction of cells by therapy. The increasing blue color intensity classifies the concentration required to achieve the same effect. The more treatment required, the more intense the blue color. DRI values represent the order of magnitude (fold) of dose reduction in a combination setting compared with each drug alone.

**Table 1 ijms-26-02829-t001:** HPV status of cervical cancer cells [42].

Genotype	HPV 16	HPV 18
SiHa	**+**	**-**
CaSki	**+**	**-**
C-33A	**-**	**-**

## Data Availability

The data presented in this study are available on request from the corresponding author.

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
