# Peer review of "Dual Topoisomerase Inhibitor Is Highly Potent and Improves Antitumor Response to Radiotherapy in Cervical Carcinomaâ€"

_ijms, 2025, doi:10.3390/ijms26072829_

Round 1
Reviewer 1 Report
Comments and Suggestions for Authors
The authors have prevented a study of a compound, P8-D6, that is under development as a dual topoisomerase I/II poison. This compound presents a unique opportunity for cancer therapy with the ability to poison both topo I and topo II, both of which are known, effective anticancer drug targets.
Overall, the manuscript is well-written and clear. The authors present the data clearly and fairly. They also provide adequate access to the raw data files and images.
In terms of questions, there is a minor point that could be clarified. The authors utilize cisplatin as the comparison given that this is a drug used to treat cervical carcinoma. However, the authors do not show whether this compound is superior to either topoisomerase I or II poisons, such as topotecan or etoposide. Why are there no comparisons with either of these compounds either individually or in combination? Perhaps this has been shown elsewhere? Is it possible that these individual topoisomerase poisons could show similar results in combination? This does not require repeating these experiments in totality, but there needs to be some discussion here about any evidence relating to these comparisons.
Author Response
Thank you for the opportunity to resubmit our manuscript after carefully answering the comments of the reviewers. First of all, we would like to thank all reviewers for their time and great input, comprehensive recommendations and helpful suggestions improving the scientific value of our manuscript.
We have added our Point-by-point response to reviewers’ comments in the attached document/file. All changes to the manuscript are highlighted in yellow.

Reviewer 2 Report
Comments and Suggestions for Authors
This manuscript has scientific merit and may be considered for publication in the Int. J. Mol. Sci. after clarification of certain issues.
- Introduction
The structure of compound P8-D6 and the synthesis scheme of this compound should be provided because the Authors emphasize the simplicity of its synthesis.
- Results
It is difficult to follow the text without proper references to sources, for example:
Page 3, lines: 133, 135, and 147.
Page 5, lines: 166, 169, and 182.
Page 6, lines: 188, 189, and 193.
Page 7, lines: 200, 213, 214, 216, and 217.
Page 8, lines: 230, 232, 233, 236, 237, 239, and 240.
Page 9, lines: 260, 261, and 262.
Please complete.
- Discussion
1. Similarly, references to sources have been missed, for example:
Page 10, line 311.
Page 11, lines 328-331: Statement “During treatment, a significantly higher cytotoxic effect and apoptosis induction was observed with P8-D6 than with the standard therapeutic cisplatin, both in 2D and 3D cell cultures. In particular, a considerably stronger antiproliferative effect was detected compared to standards at lower concentrations” should be supported by a reference to the source.
- Page 11, lines 333-335: Sentences "Comparison with other studies showed that the results are consistent and that C-33A is also the most sensitive to cisplatin in 2D cell culture [42,84,85]. Comparison with other studies showed consistency in our results for cisplatin. C-33A cells in 2D culture were also the most sensitive to cisplatin in these studies" are confusing. It is not clear to me why the authors discuss the results of cisplatin. This section should include a discussion of the results in the context of the P8-D6 used. Please clarify.
I recommend a major revision.
Author Response

(The authors gave the same response as above.)

Reviewer 3 Report
Comments and Suggestions for Authors
The article demonstrates the efficacy of P8-D6 as a radiosensitizer using established cervical cancer cell lines and primary cell 2D/3D cultures. There are major and minor suggestions:
Major Concern: In terms of clinical translation, are the doses and radiation parameters used in this study practical for clinical application? The authors are encouraged to discuss the feasibility and relevance of these conditions in a clinical setting.
Minor Comments:
- (Method) Was cisplatin dissolved in PBS or DMSO? Please clarify.
- What is the expected major side effect of P8-D6? Additional discussion on potential toxicity would be beneficial.
There are errors in the text labeled as "Error! places." Please correct these issues
Author Response

(The authors gave the same response as above.)

Round 2
Reviewer 1 Report
Comments and Suggestions for Authors
Thank you for your response to the main concerns. One minor note below needs to be addressed.
Please correct the following:
P8-D6 was synthesized as recently described [15] (Figure S 6)and solved in PBS. 359
Cisplatin and topotecan was obtained from the UKSH dispensary and solved in PBS.
The term should be "dissolved" rather than "solved".
Reviewer 2 Report
Comments and Suggestions for Authors
I would like to thank the Authors for incorporating my comments in the revised manuscript. As a result, I can accept the revised form of the submitted manuscript for publication in IJMS.
Reviewer 3 Report
Comments and Suggestions for Authors
The authors made appropriate revisions.
Author Response
Dear Reviewer,
thanks for your comment und your help in improving our manuscript.
Best regards
Inken Flörkemeier